# Impact of Probiotic Fermentation on the Physicochemical Properties of Hemp Seed Protein Gels

**DOI:** 10.3390/polym16213032

**Published:** 2024-10-29

**Authors:** Yipeng Liu, Yingxue Fei, Chen Li, Jianming Cheng, Feng Xue

**Affiliations:** 1School of Pharmacy, Nanjing University of Chinese Medicine, Nanjing 210023, China; nzylyp@163.com (Y.L.); feiyingxue0210@163.com (Y.F.); cjm@njucm.edu.cn (J.C.); 2Jiangsu Province Engineering Research Center of Classical Prescription, Nanjing 210023, China; 3College of Food Science and Light Industry, Nanjing Tech University, Nanjing 211816, China; lichenfs@njtech.edu.cn

**Keywords:** hemp seed protein isolates, plant-based yogurt, fermentation time, structure, molecular interactions

## Abstract

Hemp seed protein isolates (HPI) were used to produce a gel through probiotic fermentation. This study assessed how fermentation time (ranging from 0 to 16 h) affected the physicochemical properties of the HPI gel. The results indicated that gel formation began after 8 h of fermentation, as demonstrated by a pH decrease, an increase in particle size, and the development of aggregation observed through fluorescence and scanning electron microscopy. The gel produced after 16 h of fermentation showed the highest viscosity, storage modulus, and gel strength, attributed to stronger molecular interactions, including non-covalent and covalent crosslinking. However, the gel produced after 12 h of fermentation showed the highest water-holding capacity, and extending the fermentation beyond 12 h caused a decrease in water-holding capacity. Additionally, the subunits tended to form polymers after fermentation, suggesting that gel formation was influenced by both acidification and specific covalent crosslinking. These findings propose that HPI could serve as a viable alternative for developing plant-based gel products.

## 1. Introduction

By 2050, the global population is projected to reach 9.7 billion, significantly increasing the demand for food, especially protein, which is crucial for human health [1]. Currently, animal protein is the primary source of dietary protein, but its production is associated with low energy efficiency, high costs, substantial environmental impact, and potential health risks [2]. Consequently, plant-based protein foods offer a more sustainable alternative. Recently, the plant-based food industry has grown rapidly, with expectations that the global market for plant-based foods will reach USD 162 billion by 2030 [3]. As a result, identifying renewable and sustainable alternatives to animal-based proteins has become a top priority.

Plant-based foods generally fall into three categories: plant-based meat, plant-based milk, and plant-based eggs. Among these, the plant-based yogurt market shows significant promise, with projections indicating it could reach USD 144 billion globally by 2029 [4,5]. Currently, the primary ingredients for plant-based yogurt include bulk agricultural products like soy, oat, pea, and rice proteins. Additionally, medicinal and edible proteins, such as those found in hemp seed, plum seed, and jujube seed, which each contain abundant proteins, offer valuable alternatives [6]. These proteins not only serve as potential raw materials for plant-based yogurt but also possess specific physiological benefits.

Hemp seed, a versatile crop with medicinal and culinary uses, originates from China and has been consumed for over 3000 years. It is commonly cultivated in regions like Guangxi, Yunnan, Liaoning, Jilin, and Sichuan provinces. Hemp seeds are highly nutritious, containing about 25% protein [7]. Compared to other plant proteins, hemp seed protein offers distinct benefits: it closely resembles human protein, making it particularly suitable for human dietary needs [8]; its amino acid profile is well-balanced and meets the ideal standards set by the Food and Agriculture Organization, thus being recognized as a “high-quality complete protein” by nutritionists [9]; it does not contain tryptophan inhibitors, which improves digestibility and absorption [9]; it has a mild, fresh taste with no bean-like odor [10]; and it positively affects blood pressure, blood sugar levels, digestion, weight management, and immune regulation, while also offering anti-tumor properties and alleviating symptoms of anemia, hypoxia, and fatigue [11,12]. Despite its health benefits, hemp seed protein is often underutilized because hemp seed meal, a by-product of oil extraction, is commonly used as animal feed rather than being harnessed as a valuable protein resource.

As a polymer, hemp seed protein is also capable of forming a gel through intermolecular interactions. Currently, research on hemp seed protein gel primarily centers around heat-induced gelation [13]. In recent times, there has been significant interest in utilizing probiotics fermentation to induce gel formation in plant proteins. The gel formation mechanism primarily involves the use of probiotics to produce lactic acid through fermentation, thereby lowering the system’s pH and ultimately bringing it closer to the protein’s isoelectric point, thus inducing gel formation [14]. In other words, slow acidification is the primary mechanism by which probiotics induce gel formation through fermentation. However, our recent study on plum seed protein revealed that partial protein hydrolysis, induced by proteases secreted by probiotics, plays a role in the formation of plum seed protein gel [15]. Nevertheless, the mechanism by which probiotics induce gel formation in hemp protein remains unclear.

Therefore, the goal of this study was to create plant-based yogurt using hemp seed protein isolates (HPI) as a gelling agent. To understand the gelation mechanism, we investigated how fermentation time (ranging from 0 to 16 h) impacts the physicochemical properties, rheological properties, structure, and molecular interactions of HPI. This research aims to broaden the application of HPI as an innovative protein source in the food industry.

## 2. Materials and Methods

### 2.1. Materials and Chemicals

Hemp seeds were sourced from Shaanxi Chuliang Agricultural Technology Co., Ltd. (Shaanxi, China) and ground using a Breville coffee grinder (Model BCG 300, Breville, Sydney, Australia). The hemp seed powder was defatted with a mixture of n-hexane and ethanol (10:1, *v*/*v*), and hemp seed protein isolates (HPI) were extracted following the method described in our previous study [16]. Before the protein solution was freeze-dried, ultrasound (400 W, 12 min) was applied to treat the protein solution to increase its solubility. The hydrophobic fluorescent probe 8-Anilino-1-naphthalenesulfonic acid (ANS) and the fluorescence dye fluorescein isothiocyanate (FITC) were obtained from Sigma-Aldrich Chemical Company (St. Louis, MO, USA) and Shanghai Yuanye Biotechnology (Shanghai, China), respectively. Starter cultures, which included *Lactobacillus bulgaricus*, *Streptococcus thermophilus*, *Lactobacillus acidophilus*, *Lactobacillus plantarum*, and *Lactobacillus casei*, were purchased from Kunshan Baishengyou Biotechnology (Suzhou, China). Other reagents were procured from Sinopharm Chemical Reagent Co., Ltd. (Shanghai, China).

### 2.2. Preparation of PSPI-Based Gel

The 8% hemp seed protein solution was prepared by combining HPI with pure water, followed by the dissolution of 5% sugar into the mixture. The solution was stirred thoroughly and kept in the refrigerator overnight. The next day, the mixture was homogenized using a FJ-200 high-speed disperser (Shanghai Specimen Model Factory, Shanghai, China) to ensure uniformity. The solution was then heated in boiling water for 20 min to alter the protein properties, followed by rapid cooling in cold water to 20 °C. A 1% starter culture was added to the solution, which was then mixed well and transferred into 150 mL bottles. The bottles were placed in a fermentation box set to 42 °C for fermentation periods of 0, 4, 8, 12, and 16 h. After fermentation, portions of the samples were used to evaluate gel properties, while the remaining portions were poured into Petri dishes, pre-frozen, and freeze-dried to assess the functional characteristics of the gel powder.

### 2.3. Effects of Fermentation on Physicochemical Properties of PSPI Gel

#### 2.3.1. Effects of Fermentation on pH Value and Particle Size

The gel sample was homogenized by stirring at 600 rpm for 5 min using an MS5S blender (Qunan Experimental Instrument Co., Ltd., Huzhou, China), and the pH was measured with a PH-100 pH meter (Shanghai Lichen Bangxi Instrument Technology Co., Ltd., Shanghai, China). Additionally, the gel sample was dispersed in deionized water at a 1% (*m*/*v*) concentration. Particle size distribution was analyzed using light scattering with an LS13320 laser particle size analyzer from Beckman Coulter Trading (Shanghai, China) Co., Ltd.

#### 2.3.2. Effects of Fermentation on Rheological Behavior

All rheological tests were carried out on gels that had been refrigerated at 4 °C for 24 h without any stirring post-fermentation. For each test, 2 g of the gel were carefully placed onto the rheometer plate [17]. Measurements were taken using a rheometer (HR-1, TA Instruments, Leatherhead, UK) with a parallel plate configuration, consisting of a 60 mm diameter plate and a 100 μm gap. Initially, a continuous shear test was conducted to assess viscosity across a shear rate range of 0.1 to 1000 s^−1^. Following this, a dynamic oscillatory shear test was performed to measure the storage and loss moduli over an angular frequency range of 0.1 to 100 rad/s at a fixed strain amplitude of 0.1%. Finally, creep and recovery tests were carried out at a pressure of 5 Pa and a temperature of 25 °C, with a creep duration of 300 s and a recovery duration of 300 s. The relative recovery rate was calculated [18].

#### 2.3.3. Effects of Fermentation on Water Holding Capacity and Gel Strength

The gel sample was subjected to centrifugation at 10,000× *g* for 10 min using a centrifuge from Beijing Dalong Xingchuang Experimental Instrument Co., Ltd., Beijing, China, and the supernatant was then discarded. The water holding capacity (WHC) was determined as the percentage of water retained in the gel after centrifugation. To evaluate the gel’s strength, 100 mL of the gel, held in a glass cylindrical container (50 mm diameter), was tested with a Rapid TA texture analyzer (Shanghai Tengba Instrument Technology Co., Ltd., Shanghai, China). The maximum force was recorded as the gel was penetrated at a constant speed of 1 mm/s to a 50% compression strain using a stainless-steel cylinder probe (36 mm diameter).

#### 2.3.4. Effects of Fermentation on Microstructure

To detect protein components in the gel samples, 1 g of fresh gel was combined with 10 μL of fluorescein isothiocyanate (0.1 mg/mL in dimethyl sulfoxide) and analyzed with a fluorescence microscope (DM2500, Leica, Wetzlar, Germany). For a more detailed examination of the microstructure, the fresh gel was first lyophilized. The resulting sample was then coated with gold by spraying and observed using a scanning electron microscope (SU8010, Hitachi, Japan).

#### 2.3.5. Effects of Fermentation on Molecular Interactions

The intrinsic fluorescence emission spectra of tryptophan (Trp) in lyophilized gel samples (1 mg/mL) were measured between 300 and 600 nm with an excitation wavelength set at 285 nm, utilizing a Spark 10M spectrophotometer (Tecan, Männedorf, Switzerland). Fluorescence intensity for sample dispersions (0.05 to 2 mg/mL, with 8 mM ANS reagent) was recorded at an excitation wavelength of 390 nm and an emission wavelength of 470 nm. Surface hydrophobicity (H_0_) was determined according to established procedures. The content of sulfhydryl (SH) groups was evaluated by combining 3 mL of a 2 mg/mL sample dispersion with 0.03 mL of a 4 mg/mL DTNB reagent and measuring absorbance at 412 nm [19]. Free amino group content was measured by reacting 200 μL of a 1 mg/mL sample dispersion with 4 mL of OPA reagent at 90 °C for 5 min, with absorbance at 340 nm recorded using a UV spectrophotometer (UV-5500, METASH, China) and quantification based on a lysine standard curve [15]. For solubility testing, 0.5 g of lyophilized gel was dissolved in 10 mL of different solvents: DW (deionized water at pH 8.0), B (Tris-glycine buffer: 0.086 M Tris, 0.09 M glycine, and 4 mM Na_2_EDTA, pH 8.0), and BSU (Buffer B with 0.5% SDS and 8 M urea) [20]. The dispersions were initially homogenized at 10,000 rpm for 1 min using a homogenizer (FJ-200, Shanghai Specimen Model Factory, China), followed by centrifugation at 10,000× *g* for 30 min at 4 °C. Protein concentration in the supernatant was assessed using the Lowry method with a BSA standard curve [21].

#### 2.3.6. Effects of Fermentation on Subunits

The subunits of the samples were examined using sodium dodecyl sulfate–polyacrylamide gel electrophoresis (SDS-PAGE). Lyophilized samples were reconstituted in loading buffer. A 10 μL aliquot of each sample dispersion was applied to the gel, and electrophoresis was conducted at 200 V for 30 min using the Mini-Protein Tetra Electrophoresis System (Bio-Rad Laboratories, Inc., Hercules, CA, USA) [22].

### 2.4. Statistical Analysis

All experiments were conducted in triplicate, and the results were presented as mean ± standard deviation. To identify significant differences (*p* < 0.05), one-way analysis of variance was used, followed by Duncan’s multiple range test. The statistical analyses were performed using SPSS for Windows version 17.0 (IBM, USA).

## 3. Results and Discussion

### 3.1. Appearance of Gel and pH Value

As illustrated in Figure 1, probiotic fermentation led to gel formation in HPI, with the extent of gelation depending on the fermentation duration. Gel formation began after 8 h of fermentation and was complete by 12 h. Previous studies showed that the gel of soy proteins was complete by 15 h of fermentation [14] and that the gel of plum seed proteins was complete by 9 h of fermentation [15]. However, after 16 h of fermentation, syneresis (red circle) was observed, suggesting that extended fermentation beyond this point could compromise the gel structure.

Figure 2 shows that the pH value decreased with longer fermentation times, which might be linked to reduced surface charge leading to protein aggregation and gel formation. Plant proteins typically have positive surface charges, and as the pH drops, these positive charges diminish, reducing electrostatic repulsion and promoting aggregation. This phenomenon has been previously documented in studies of acid-induced protein gels, in which protein particles aggregated during acidification due to charge neutralization [23]. Earlier research indicated that HPI had an isoelectric point between pH 4.3 and 6.0 [9]. Therefore, after 4 h of fermentation, the pH approached the isoelectric point of PSPI, suggesting that gel formation occurred through gradual acidification. Previous research on plant-based yogurts, including those from soy protein and plum seed protein, also showed that organic acids produced during probiotic fermentation were key to inducing protein gelation [1,14]. Additionally, the pH value dropped to 4.4 after 12 h of fermentation. This result was similar to that in a previous study, in which hemp-protein-based-yogurt possessed a pH value of 4.5 [24]. Our result also aligned with the pH values of other plant-protein-based yogurts. For instance, pea-protein- and mung-bean-protein-based yogurts typically had a pH of 4.6 [25,26]; soy-protein-based yogurts had a pH value of 4.5 [27]; black soybean yogurt had a pH of 4.7 [28]; and the pH values for soymilk yogurts were in the range of 4.4–4.7 [29] or below 5.0 [30].

### 3.2. Particle Size

As depicted in Figure 3, probiotic fermentation for 4 h had no significant impact on the particle size of the HPI. However, after 8 h of fermentation, the particle size of the HPI began to increase, reaching its peak at 16 h. This increase in particle size was likely due to the aggregation of HPI induced by fermentation. This finding aligned with the observation that the HPI gelation commenced after 8 h and was fully developed by 12 h of fermentation. Additionally, the particle size of the HPI-based yogurt was notably larger compared to plum-seed-protein-based yogurt [15], soy-protein-based yogurt [9], and pea-protein-based yogurt [31], suggesting that the HPI tended to form larger aggregates during fermentation.

### 3.3. Rheological Behavior

Figure 4 illustrates that the viscosity of the samples decreased as the shear rate increased, indicating that all the samples exhibited shear-thinning behavior [32]. This behavior suggested that as the shear rate increased, the interactions among the HPI molecules were disrupted [5]. This result was consistent with those of a precious study about soy-protein-based yogurt, in which the viscosity increased with extension of the fermentation time [14]. Additionally, the viscosity of the HPI-based yogurt was comparable to that of soy protein-based yogurt [18], but was lower than those of fava beans, chickpea-based yogurts, and plum-seed-protein-based yogurt [33].

As shown in Figure 5 and Figure 6, the storage modulus (G′) of the HPI-based yogurt was higher than the corresponding loss modulus (G″), suggesting that all the samples showed “gel” like behavior. The previous studies about coconut-milk-based, soy-protein-based, and soy-milk-based yogurts also showed the similar results [14,34,35]. Additionally, the storage modulus of the HPI-based yogurt was lower than those of plum-seed-protein-based [3], pea-protein-based [26], soy-protein-based [14], chickpea-, and lentil-protein-based yogurt [36], indicating that the HPI-based yogurt likely had a weaker internal structure.

The creep recovery test is a method that can be employed to investigate the viscoelastic properties of gel. Typically, a higher strain value indicates a weaker gel structure [37]. As depicted in Figure 7, probiotic fermentation decreased the value of the strain, suggesting an increase in the elasticity and strengthening of HPI-based yogurt. Furthermore, the recovery rates for the HPI-based yogurt from 0 to 16 h of fermentation were 52.04 ± 1.02, 57.32 ± 2.11, 48.15 ± 2.35, 66.69 ± 1.84, 43.47 ± 3.05, respectively. This result indicated that the HPI-based yogurt from 12 h of fermentation exhibited the best resistance to deformation [14].

### 3.4. Water-Holding Capacity (WHC) and Strength of Gel

As depicted in Figure 8, the water-holding capacity (WHC) of the HPI improved significantly once gel formation began after 4 h of fermentation. This increase in WHC was likely due to the development of a network structure through intramolecular cross-linking or polymerization of the HPI, which could have helped to retain water against centrifugal forces. The highest WHC value was obtained after 12 h of fermentation but declined with further fermentation beyond this time. Previous studies indicated that the WHC of protein-based gels was affected by their microstructure and intramolecular interactions [38,39]. Typically, gels that have more uniform network structures and stronger intramolecular bonds exhibit higher WHC. Our findings aligned with this understanding because the HPI gel fermented for 12 h showed enhanced WHC due to stronger intramolecular interactions. The reduction in WHC after 12 h might have been due to the formation of a more compact, non-porous gel structure, which decreased its water-holding ability. A previous study about soy-protein-based yogurt revealed that syneresis occurred after excessive fermentation due to the formation of non-self-supporting gel [40]. Additionally, the WHC of the HPI-based yogurt was comparable to that of plum-seed-based [15] and soybean-based yogurt [41].

As shown in Figure 9, the gel strength of the HPI improved significantly once gel formation began after 4 h of fermentation. The gel strength augmented as the fermentation time elongated, further validating the observation that the molecular interactions in the gel intensified with prolonged fermentation time. Similar results were also observed in soy-protein- [14] and plum-seed-protein-based yogurt [15].

### 3.5. Microstructure of Gel

The gel formation during fermentation was examined using fluorescence microscopy. As illustrated in Figure 10, the HPI began to aggregate after 8 h of fermentation, as was evident from the appearance of larger green aggregates and the increase in fluorescence intensity. With an extended fermentation time (from 8 to 16 h), these aggregates became larger and denser, indicating the progression of gel formation. A previous study about plum seed protein-based gel also found that proteins tended to form aggregates with the extension of the fermentation time [15].

The microstructure of the samples fermented for 8 to 16 h was also analyzed using scanning electron microscopy (SEM), with the results shown in Figure 11. After 8 h of fermentation, the HPI gel showed a discontinuous network structure. At 12 h of fermentation, pores began to form on the surface of the HPI gel, indicating gel development. Similar porous structures have been reported in studies of pea-protein-based [17], soy-protein-based [14], and plum-seed-protein-based yogurt [15]. However, the porous structure disappeared and a more compact structure was observed on the surface of the HPI gel after 16 h of fermentation. A previous study about soy-protein-based yogurt showed that the porous structure disappeared with excessive fermentation due to enhanced molecular forces [14]. This result also explained why the gel, from 12 h of fermentation, exhibited a higher water-holding capacity (WHC), and why the WHC decreased as the fermentation time was extended to 16 h. Previous studies also demonstrated that gels with more uniform distributions of pores and homogeneous structures exhibited better ability to hold water [42,43].

### 3.6. Molecular Interactions

Intrinsic fluorescence is a classic method for investigating changes in THE tertiary structure of protein [44]. As illustrated in Figure 12, fermentation led to a reduction in fluorescence intensity for all THE samples. These changes in intrinsic fluorescence were likely due to the formation of HPI aggregates, which could have obscured the fluorescence and reduced the intensity. The decrease in fluorescence intensity also suggested the formation of aggregates [22], which was also observed in the microstructural analysis and SEM. Previous studies showed that interactions between proteins and polysaccharides could similarly diminish fluorescence due to the shielding effect of polysaccharides [45,46,47]. A previous study also proved that the fluorescence intensity of protein in the aggregated form was lower than that in the unfolded state [48]. Our results were also consistent with the studies about soy-protein-based and plum-seed-protein-based yogurt [14,15].

As shown in Figure 13, fermentation decreased the surface hydrophobicity of the HPI. This result was consistent with the study about soy-protein-based gel, which the author of that study attributed to the decrease in surface hydrophobicity to the formation of protein aggregates [14]. The decrease in the surface hydrophobicity also indicated that hydrophobic interaction might be one of the intermolecular forces that induce HPI to form gel during fermentation, which could be attributed to the unfolding of protein induced by fermentation, followed by the exposure of more hydrophobic groups involved in the formation of the gel [49]. The previous study about pea-protein-based yogurt also found that hydrophobic interaction was the main force inducing the formation of gel [50]. The study about soy-protein-based gel also reported that the hydrophobic interaction was involved in gel formation [51]. However, the study about plum-seed-protein-based gel reported that the fermentation increased the surface hydrophobicity [15]. The inconsistent results might have been due to the occurrence of partial hydrolysis induced by fermentation, which led to the exposure of the hydrophobic groups.

It is well known that disulfide (S-S) bonds are crucial for gel formation, influencing the physicochemical properties and microstructure of gels [52]. Therefore, we measured the free SH content of the HPI during fermentation. As shown in Figure 14, fermentation led to a decrease in the free SH content, likely due to oxidation and the formation of protein–S-S–protein bonds [53]. This decrease indicates that S-S bonds contribute to gel formation. The studies about soy-protein-based [54], myofibrillar-protein-based [55], oat-protein-based [56], and pea protein-based [57] gels proved that disulfide bonds were incorporated into the gel network structure. In addition, our previous study about plum-seed-protein-based yogurt also showed that fermentation decreased the free SH content due to the formation of S-S bonds [15].

To investigate non-covalent molecular interactions, solubility of sample in various buffers was analyzed [58]. Buffer B was used to disrupt electrostatic interactions, while buffer BSU targeted hydrogen bonds and hydrophobic interactions [20]. As shown in Figure 15, the solubility of sample in buffers decreased with increasing fermentation time, indicating that gel formation led to the appearance of insoluble aggregates. For each sample, solubility in buffer B was higher than in DW, indicating that electrostatic interactions were significant in the HPI gel network. In addition, solubility in buffer BSU was much greater than in buffer B, highlighting the importance of hydrogen bonds and hydrophobic interactions in the gel network. Previous studies about plum seed protein-based and pea protein-based yogurt also found that the gel was stabilized by non-covalent interactions, such as hydrogen bonds and hydrophobic interactions [15,25]. This finding aligns with the observation on surface hydrophobicity, indicating that probiotic fermentation reduced the surface hydrophobicity of HPI. Consequently, proteins assembled into gels via hydrophobic interactions. When urea in the buffer BSU disrupted the hydrophobic interactions among protein molecules. It decreased protein aggregation, thereby enhancing protein solubility.

### 3.7. Subunits Analysis

As depicted in Figure 16, fermentation led to the polymerization of subunits between 55–170 kDa of HPI. This observation contrasts with previous research on lupin oat-based yogurt, which found that fermentation did not affect subunit profile of protein [59]. Similar findings were also reported for rice protein-based [60] and soy protein-based yogurts [14], where protein subunits were not affected by fermentation. In this study, the polymerization of subunits might have been caused by proteolytic enzymes from probiotics, which could induce the formation of covalent crosslinking (i.e., non-disulfide bonds). Previous studies have reported that microbial enzymes (such as transglutaminase) could induce the covalent crosslinking among protein subunits [61,62], which was also used to enhance the gel texture of peanut milk yogurt [63]. We hypothesize that the enzyme-catalyzed cross-linking process might also take place among the amino groups of amino acids. This potential mechanism was illustrated in Figure 17. Unlike these cases of broad enzymatic crosslinking, the polymerization observed in HPI was specific, targeting only subunits of HPI. Future studies should focus on isolating this specific protease and elucidating its mechanism for crosslinking of HPI. Additionally, this crosslinking might contribute to the enhanced molecular interactions and compact structure during fermentation. Thus, HPI gels form not only through gradual acidification but also through special covalent crosslinking.

## 4. Conclusions

In this study, we demonstrated that HPI can form a gel via probiotic fermentation, establishing its potential as a promising alternative in the realm of plant-based yogurts. Our results revealed that the characteristics of the HPI gel are influenced by fermentation duration, with the most favorable outcomes observed after 12 h. The gelation mechanism encompasses both progressive acidification and unique covalent crosslinking, stabilized by essential factors such as disulfide bonds, hydrogen bonds, hydrophobic interactions, and subunit polymerization. These findings underscore the practical applicability of HPI-based yogurt, particularly within the food industry and plant-based protein markets. To further enhance the impact of this study, future research endeavors should concentrate on delving into the nutritional advantages, health benefits, and flavor attributes of HPI-based yogurt. This could involve comprehensive nutritional profiling, clinical trials to assess health impacts, and the development of innovative flavoring techniques to cater to diverse consumer preferences. Additionally, exploring the scalability of the production process and assessing the shelf-life stability of HPI-based yogurt would be crucial for its successful commercialization. Such research directions not only broaden the understanding of HPI’s potential but also pave the way for groundbreaking innovations in the plant-based protein market, promoting healthier and more sustainable food choices.

## Figures and Tables

**Figure 1 polymers-16-03032-f001:**
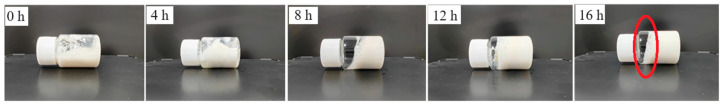
Photographs Illustrating Hemp Seed Protein Isolate (HPI) Gel Formation Over a 0–16 h Fermentation Period. The Red Circle Represents Syneresis.

**Figure 2 polymers-16-03032-f002:**
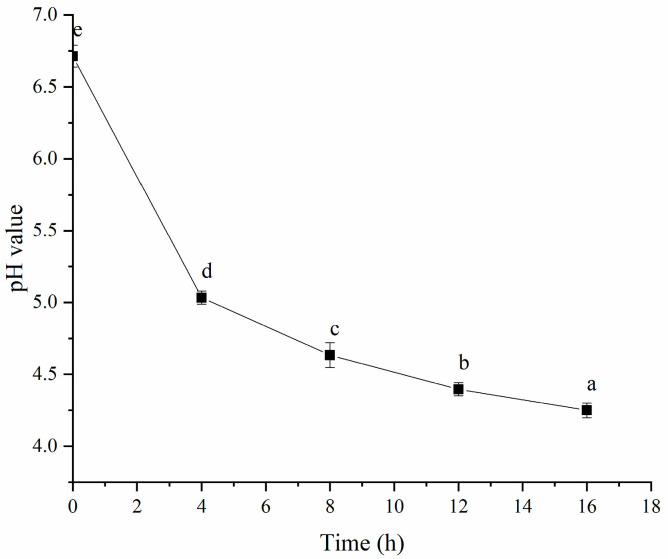
Effects of Fermentation Time (0–16 h) on pH Value of Hemp Seed Protein Isolate (PSPI) Gel. Different Letters Represent Significant Differences (*p* < 0.05).

**Figure 3 polymers-16-03032-f003:**
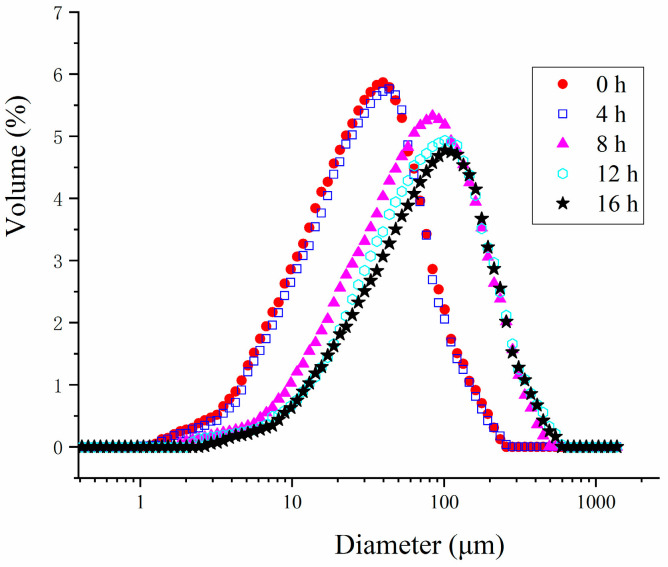
Effects of Fermentation Time (0–16 h) on Particle Size of Hemp Seed Protein Isolate (HPI) Gel.

**Figure 4 polymers-16-03032-f004:**
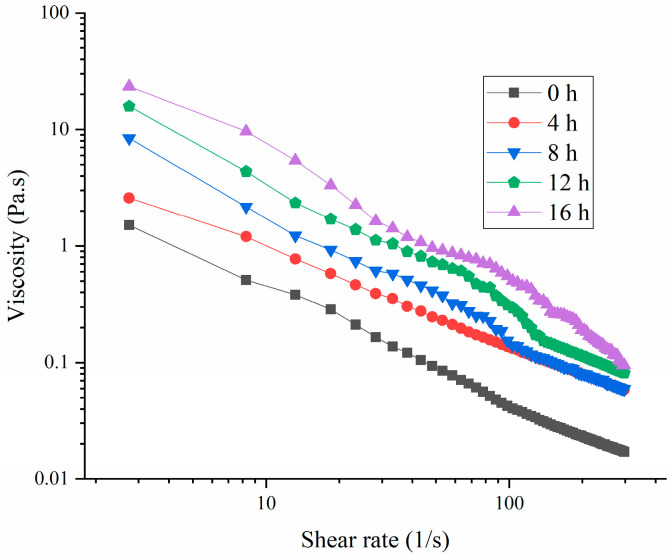
Effects of Fermentation Time (0–16 h) on Viscosity of Hemp Seed Protein Isolate (HPI) Gel.

**Figure 5 polymers-16-03032-f005:**
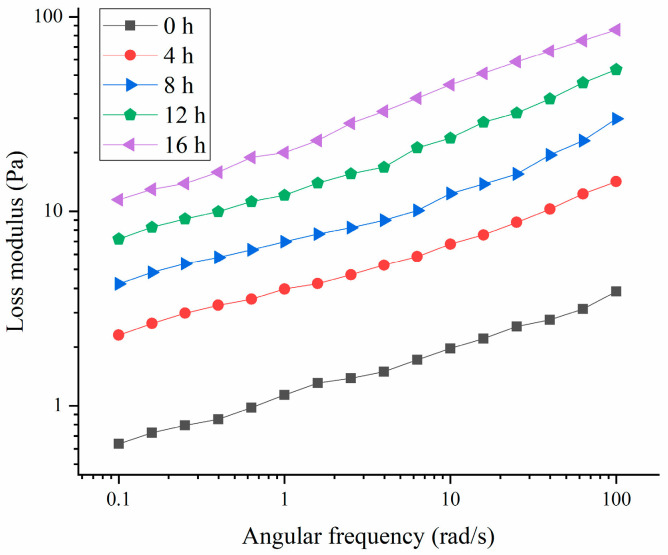
Effects of Fermentation Time (0–16 h) on Loss Modulus of Hemp Seed Protein Isolate (HPI) Gel.

**Figure 6 polymers-16-03032-f006:**
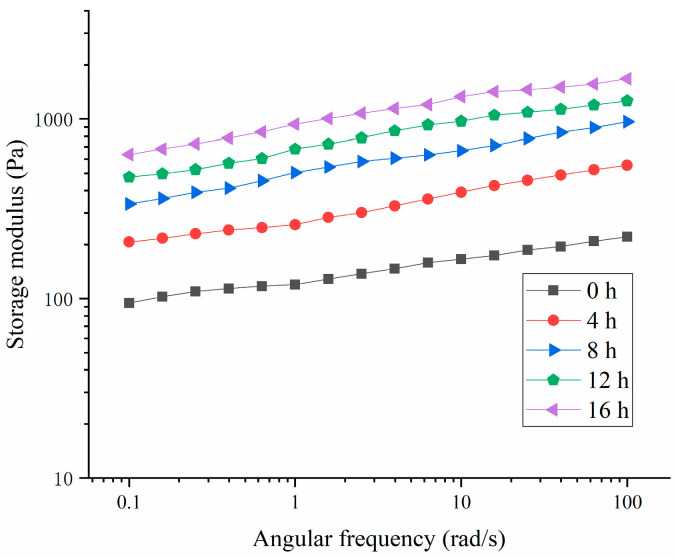
Effects of Fermentation Time (0–16 h) on Storage Modulus of Hemp Seed Protein Isolate (HPI) Gel.

**Figure 7 polymers-16-03032-f007:**
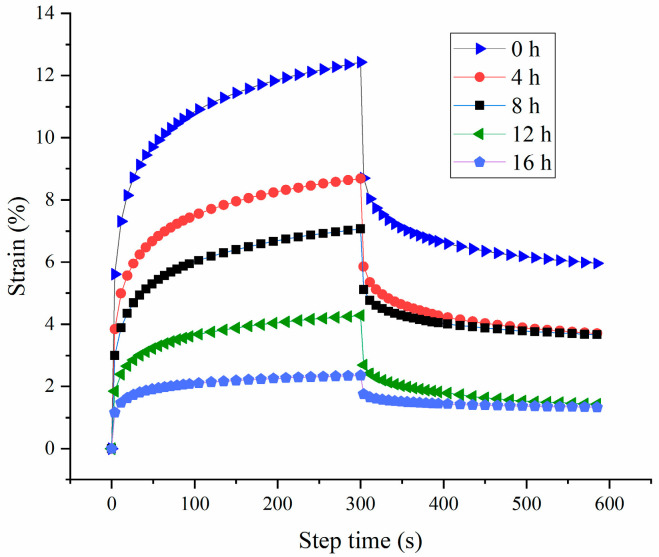
Effects of Fermentation Time (0–16 h) on Creep and Recovery Property of Hemp Seed Protein Isolate (HPI) Gel.

**Figure 8 polymers-16-03032-f008:**
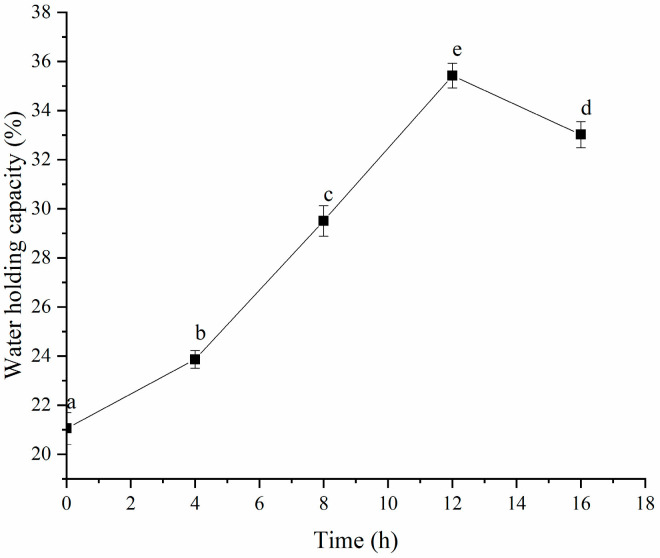
Effects of Fermentation Time (0–16 h) on Water-Holding Capacity of Hemp Seed Protein Isolate (HPI) Gel. Different Letters Represent Significant Differences (*p* < 0.05).

**Figure 9 polymers-16-03032-f009:**
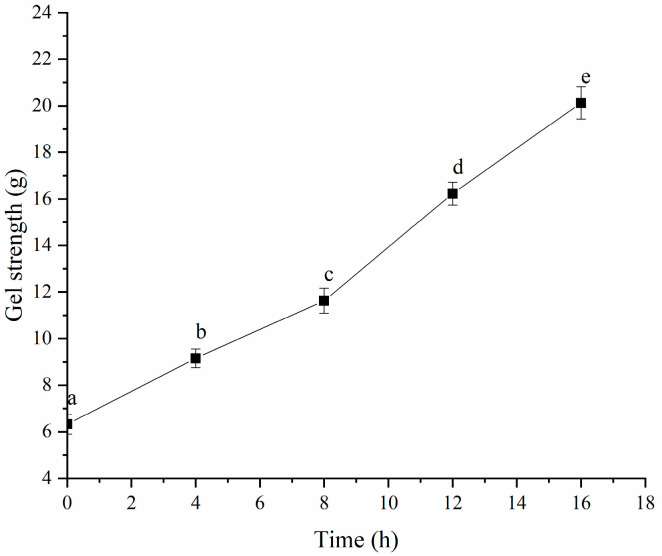
Effects of Fermentation Time (0–16 h) on Gel Strength of Hemp Seed Protein Isolate (HPI) Gel. Different Letters Represent Significant Differences (*p* < 0.05).

**Figure 10 polymers-16-03032-f010:**
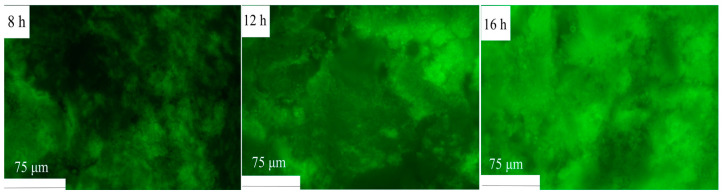
Microstructural Analysis of Hemp Seed Protein Isolate (HPI) Gel Induced by Different Fermentation Durations (8–16 h). The Green Part Represents Proteins.

**Figure 11 polymers-16-03032-f011:**
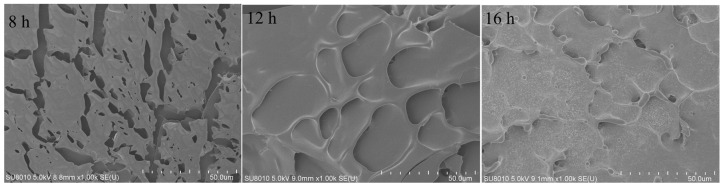
Scanning Electron Microscopy of Hemp Seed Protein Isolate (HPI) Gel Formed at Different Fermentation Times (8–16 h).

**Figure 12 polymers-16-03032-f012:**
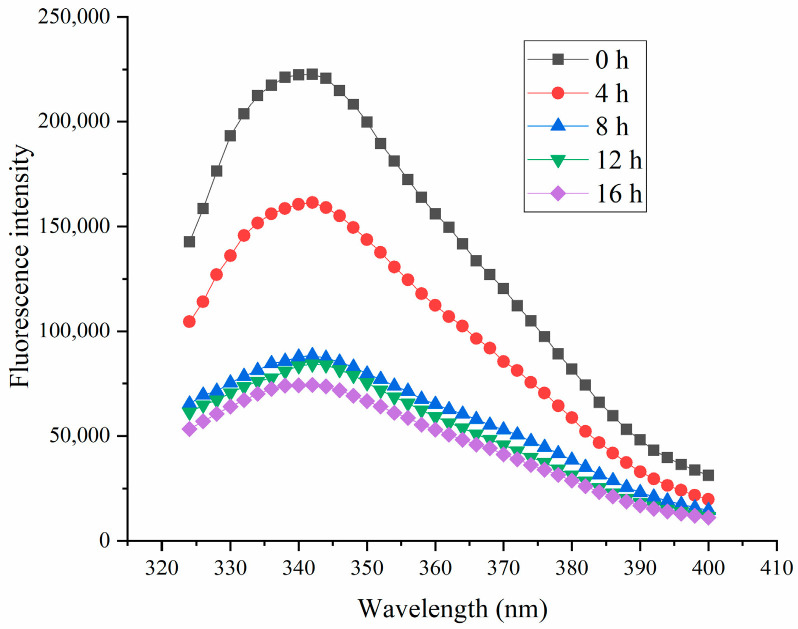
Effects of Fermentation Time (0–16 h) on Fluorescence Intensity of Hemp Seed Protein Isolate (HPI) Gel.

**Figure 13 polymers-16-03032-f013:**
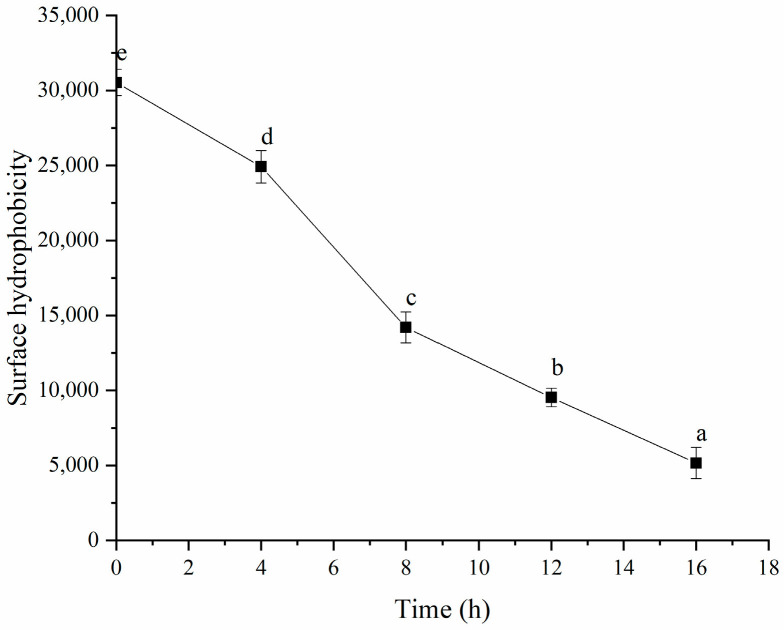
Effects of Fermentation Time (0–16 h) on Surface Hydrophobicity of Hemp Seed Protein Isolate (HPI) Gel. Different Letters Represent Significant Differences (*p* < 0.05).

**Figure 14 polymers-16-03032-f014:**
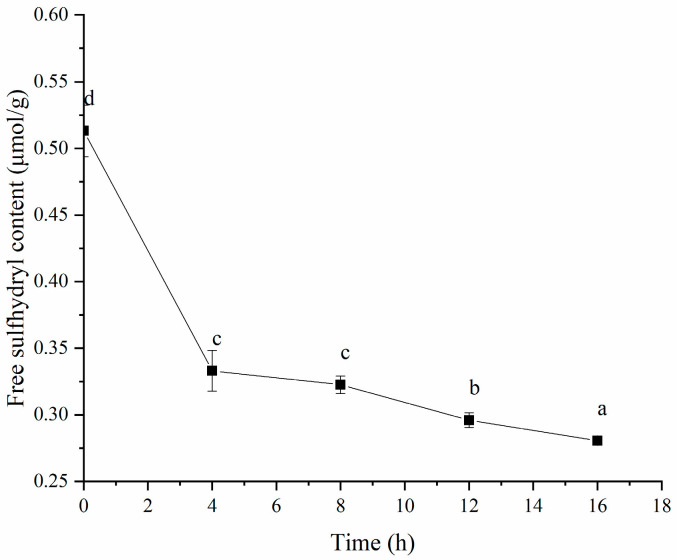
Effects of Fermentation Time (0–16 h) on Free Sulfhydryl Content of Hemp Seed Protein Isolate (HPI) Gel. Different Letters Represent Significant Differences (*p* < 0.05).

**Figure 15 polymers-16-03032-f015:**
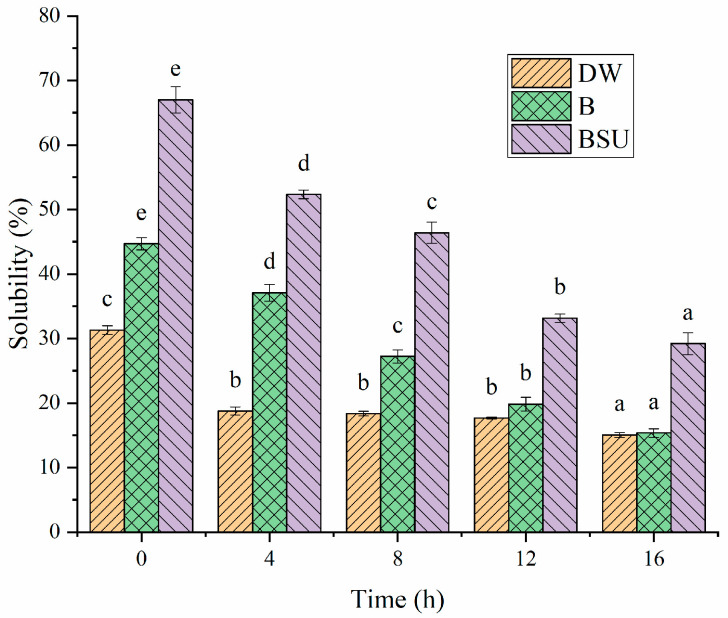
Solubility of Hemp Seed Protein Isolates (HPI) Gel Induced by Different Fermentation time (0–16 h) in Different Solvents. DW: deionized water. B: tris-glycine buffer (0.086 M tris, 0.09 M glycine,4 mM Na2EDTA). BSU: solution B (containing 0.5% SDS, 6 M urea). Different Letters in Same Pattern Represent Significant Differences (*p* < 0.05).

**Figure 16 polymers-16-03032-f016:**
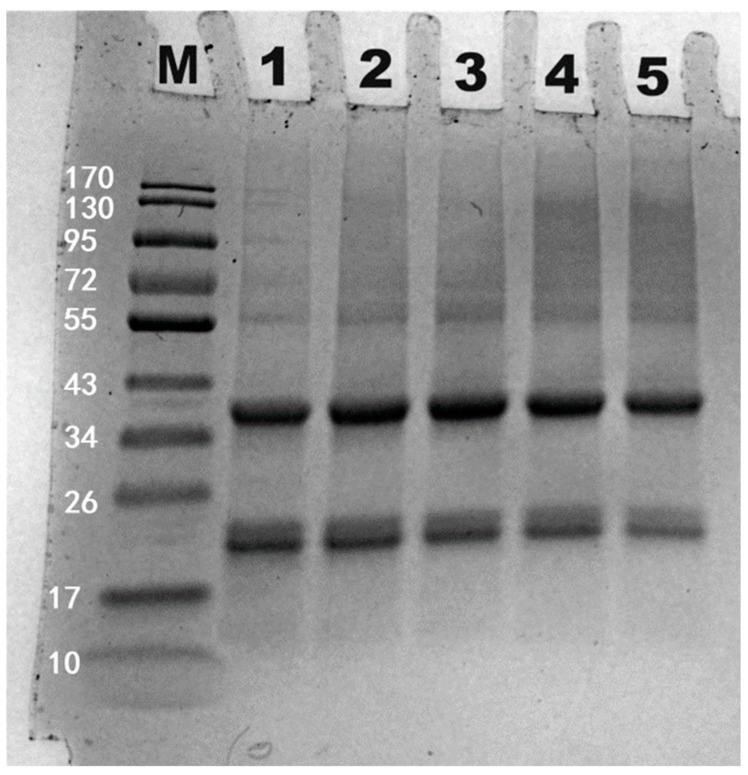
SDS-PAGE pattern of Hemp Seed Protein Isolates (HPI) Gel Induced by Different Fermentation Time (0–16 h). M: markers (kDa). 1–5: 0–16 h. The White Numbers (10–170) Represent the Molecular Weight of Markers.

**Figure 17 polymers-16-03032-f017:**
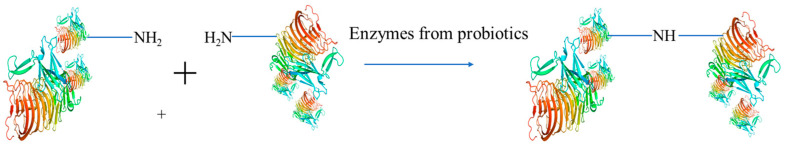
Potential Mechanism of Enzyme-catalyzed Cross-linking Process for Hemp Seed Protein Isolates (HPI) Induced by Probiotics Fermentation.

## Data Availability

All data generated or analyzed during this study are included in this manuscript.

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
