# Peer review of "Impact of Probiotic Fermentation on the Physicochemical Properties of Hemp Seed Protein Gels"

_polymers, 2024, doi:10.3390/polym16213032_

Round 1
Reviewer 1 Report
Comments and Suggestions for Authors
-
The manuscript provides valuable insights into the impact of probiotic fermentation on hemp seed protein gels, though certain sections, particularly the introduction, would benefit from improved coherence and flow, with revisions aimed at enhancing clarity and readability. The methodology is generally well-structured; however, providing more details about the specific strains of probiotics used would improve replicability and strengthen the robustness of the methods. While the results are clearly explained and contribute meaningfully to the field, a more thorough comparison with existing literature on plant-based protein fermentation could enrich the discussion, particularly by highlighting how the findings align or diverge from previous research. The figures and tables are well-designed, but some graphs could benefit from clearer labeling and more detailed captions, particularly with the inclusion of statistical significance to aid interpretation. The conclusion effectively summarizes the main findings, but it could be expanded to address the broader implications of the research, particularly in the food industry and plant-based protein markets, with suggestions for future applications and research directions adding further impact to the study.

Its need to improve.
Author Response
Comment 1: The manuscript provides valuable insights into the impact of probiotic fermentation on hemp seed protein gels, though certain sections, particularly the introduction, would benefit from improved coherence and flow, with revisions aimed at enhancing clarity and readability.
Response: We inserted a paragraph in the Introduction (Line 58-69) to elucidate that hemp seed protein, being a polymer, possesses the capacity to form gel. Furthermore, the process of probiotic fermentation prompting plant protein gelation is multifaceted, potentially encompassing gradual acidification and partial hydrolysis. Nonetheless, the underlying mechanism behind probiotic fermentation inducing hemp protein gelation remains unknown.
Comment 2: The methodology is generally well-structured; however, providing more details about the specific strains of probiotics used would improve replicability and strengthen the robustness of the methods.
Response: We have contacted the supplier, but the source of the strain is a trade secret and they are unable to provide it to consumers. Only types of probiotics can be provided. We also referred to some recently published papers about probiotic fermentation, which only provided the names of suppliers and types of probiotics. doi:https://doi.org/10.1016/j.lwt.2023.115689. doi:https://doi.org/10.1016/j.ifset.2024.103610.
Comment 3: While the results are clearly explained and contribute meaningfully to the field, a more thorough comparison with existing literature on plant-based protein fermentation could enrich the discussion, particularly by highlighting how the findings align or diverge from previous research.
Response: We have compared our results with other plant-based yogurts, such as pea protein (Line 198, 213, 234, 294, 332, 348 and 364), mung bean protein (Line 198), soy protein (Line 177, 193, 199, 212, 222, 224, 232, 235, 267, 277, 294, 297, 320, 326, 333, 347 and 380), hemp protein (Line 196), fava bean (Line 225), chickpea (Line 225 and 235), rice protein (Line 380), oat protein (Line 346 and 378), lentil protein (Line 235), black soybean (Line 200), plum seed protein (Line 177, 193, 212, 225, 234, 278, 287, 295, 321, 335, 349 and 363), and peanut milk (Line 386).
Comment 4: The figures and tables are well-designed, but some graphs could benefit from clearer labeling and more detailed captions, particularly with the inclusion of statistical significance to aid interpretation.
Response: In Figure 1, we highlight the instances of whey separation after 16 hours of fermentation using red circle. The captions of Figures 2, 8, 9, 13, 14, and 15 contain the statistical analysis results. Specifically, the caption of Figure 15 outlines the composition of each buffer solution, while the caption of Figure 10 stresses that the green portion signifies proteins.
Comment 5: The conclusion effectively summarizes the main findings, but it could be expanded to address the broader implications of the research, particularly in the food industry and plant-based protein markets, with suggestions for future applications and research directions adding further impact to the study.
Response: We have revised the conclusion (Line 402-419), emphasizing the research implications and suggesting future applications and research avenues to enhance the study's overall impact.
Comment 6: Its need to improve quality of english language.
Response: We appreciate your feedback on the language of our paper. We have carefully polished the manuscript, addressing grammatical, styling, and typographical errors.
Reviewer 2 Report
Comments and Suggestions for Authors
The manuscript “Impact of Probiotic Fermentation on The Physicochemical Properties of Hemp Seed Protein Gels” examines the physicochemical properties of vegetable protein yogurt. The novelty lies in the use of hemp to obtain the protein base. The work uses modern research methods, the text is structured and presented clearly. The references correspond to the presented research. The main question arises about the compliance of this work with the Polymers journal. The Introduction does not touch upon the physicochemistry of polymers at all, it is entirely devoted to the problems of food technology and food raw materials. All the authors' references relate to the field of "food technology and food chemistry". As for the reader's interest, here too there is a doubt. Why did the authors decide to submit the article to this journal and not to the profile? Unfortunately, I cannot recommend this work for publication in Polymers.
There are also minor comments:
1. Figures 2,8,9,13,14: The “h” should be removed from the X-axis and only the numbers should be left. Time (h) is indicated in the figure captions. The X-axis should be shifted to zero.
2. Figure 10: the authors did not indicate the scale.
3. Page 12: there is some discussion of hydrophobic interactions, but there is no detailed explanation, although the Results and Discussion sections are combined.
4. The authors mention covalent and non-covalent cross-linking. What bonds are formed during covalent interactions? What groups are involved in the formation of these bonds? Please provide a reaction scheme.
Author Response
Comment 1: The main question arises about the compliance of this work with the Polymers journal. The Introduction does not touch upon the physicochemistry of polymers at all, it is entirely devoted to the problems of food technology and food raw materials. All the authors' references relate to the field of "food technology and food chemistry". As for the reader's interest, here too there is a doubt. Why did the authors decide to submit the article to this journal and not to the profile?
Response: We have included an additional paragraph in the Introduction (Line 58-69) to underscore that hemp protein, being a type of polymer, can be utilized to create gels. Furthermore, our research revealed that the molecular mechanism behind the gel formation of hemp protein through probiotic fermentation involves the induction of protein polymerization. Consequently, we contend that our research represents an application of protein as a polymer in the food industry, aligning with the journal's scope of Polymers.
Comment 2: Figures 2,8,9,13,14: The “h” should be removed from the X-axis and only the numbers should be left. Time (h) is indicated in the figure captions. The X-axis should be shifted to zero.
Response: Thank you very much for your suggestions. We have adjusted the "h" on the X-axis for Figures 2, 8, 9, 13, and 14, and the X-axis (Figures 2, 8, 9, 13, and 14) has been repositioned to start at zero.
Comment 3: Figure 10: the authors did not indicate the scale.
Response: We have incorporated a white ruler into the design, specifically placing it in the lower left corner of Figure 10.
Comment 4: Page 12: there is some discussion of hydrophobic interactions, but there is no detailed explanation, although the Results and Discussion sections are combined.
Response: Further details concerning hydrophobic interactions have been provided in Line 365-370.
Comment 5: The authors mention covalent and non-covalent cross-linking. What bonds are formed during covalent interactions? What groups are involved in the formation of these bonds? Please provide a reaction scheme.
Response: Using the microbial transglutaminase mechanism and our experimental data, we've added a diagram depicting hemp protein cross-linking induced by proteolytic enzymes from probiotics (as shown in Fig. 17).
Reviewer 3 Report
Comments and Suggestions for Authors
The manuscript entitled “Impact of Probiotic Fermentation on The Physicochemical
Properties of Hemp Seed Protein Gels”. It is a valuable and scientifically-sound paper. The idea of presented in the manuscript is quite original and seems publishable.
The background in Introduction provides main information and coarsely explains the topic of the studies. However it contains one unclear point.
On page 2 it is stated: “...its amino acid profile is well-balanced and meets the ideal standards set by the Food and Agriculture Organization, thus being recognized as a “high-quality complete protein” by nutritionists; it does not contain tryptophan inhibitors, which improves digestibility and absorption; it has a mild, fresh taste with no bean-like odor; and it positively affects blood pressure, blood sugar levels, digestion, weight management, and immune regulation, while also offering anti-tumor properties and alleviating symptoms of anemia, hypoxia, and fatigue. Despite its health benefits, hemp seed protein is often underutilized because hemp seed meal, a by-product of oil extraction, is commonly used as animal feed rather than being harnessed as a valuable protein resource.”
The is no reference, so I suggest to address mentioned characteristics and add relevant references. In is not clear whether ref. [8-10] mentioned in the first line of the paragraph cover all of those features?
Materials and methods section is widely written and no lacks are detected. Experimental part has been well performed. Results and discussion section is abundant and properly presents the results.
Regarding all above I recommend publication of the manuscript.
Author Response
Comment 1: On page 2 it is stated: “...its amino acid profile is well-balanced and meets the ideal standards set by the Food and Agriculture Organization, thus being recognized as a “high-quality complete protein” by nutritionists; it does not contain tryptophan inhibitors, which improves digestibility and absorption; it has a mild, fresh taste with no bean-like odor; and it positively affects blood pressure, blood sugar levels, digestion, weight management, and immune regulation, while also offering anti-tumor properties and alleviating symptoms of anemia, hypoxia, and fatigue. Despite its health benefits, hemp seed protein is often underutilized because hemp seed meal, a by-product of oil extraction, is commonly used as animal feed rather than being harnessed as a valuable protein resource.” The is no reference, so I suggest to address mentioned characteristics and add relevant references. In is not clear whether ref. [8-10] mentioned in the first line of the paragraph cover all of those features?
Response: We have added corresponding references after each advantage of hemp protein as suggested (Line 48-54).
Round 2
Reviewer 2 Report
Comments and Suggestions for Authors
The authors have made changes to the text, which has improved the comprehension of the information. However, the X-axis is still not shifted to zero (Fig. 2,8,9,13,14). Instead, the authors have shifted the Y-axis. Why? Please correct.
I still believe that there are more appropriate journals for this study, but if the Editors think differently, that is their decision.
Author Response
Comment 1: The authors have made changes to the text, which has improved the comprehension of the information. However, the X-axis is still not shifted to zero (Fig. 2,8,9,13,14). Instead, the authors have shifted the Y-axis. Why? Please correct.
Response: Thank you very much for your valuable suggestion. We have made the necessary modification to the X-axis coordinate of Figure 2, 8, 9, 13, 14, ensuring that these figures now start from 0. We hope that these changes will fully meet the requirements of the reviewer this time.
Comment 2: I still believe that there are more appropriate journals for this study, but if the Editors think differently, that is their decision.
Response: Thank you for your concerning. The research presented in this article, emphasizing the use of protein to prepare gel within the context of polymers' application in the food industry, fully meets the requirements outlined in the special issue (Advanced Polymers in Food Industry II).